# NOD-TAMP: Generalizable Long-Horizon Planning with Neural Object Descriptors

**Shuo Cheng** [1], **Caelan Garrett** [*2], **Ajay Mandlekar** [*2], **Danfei Xu** [1]
[1]Georgia Institute of Technology  [2]NVIDIA Corporation

**Abstract:** Solving complex manipulation tasks in household and factory settings remains challenging due to long-horizon reasoning, fine-grained interactions, and broad object and scene diversity. Learning skills from demonstrations can be an effective strategy, but such methods often have limited generalizability beyond training data and struggle to solve long-horizon tasks. To overcome this, we propose to synergistically combine two paradigms: Neural Object Descriptors (NODs) that produce generalizable object-centric features and Task and Motion Planning (TAMP) frameworks that chain short-horizon skills to solve multi-step tasks. We introduce NOD-TAMP, a TAMP-based framework that extracts short manipulation trajectories from a handful of human demonstrations, adapts these trajectories using NOD features, and composes them to solve broad long-horizon, contact-rich tasks. NOD-TAMP solves existing manipulation benchmarks with a handful of demonstrations andsignificantly outperforms prior NOD-based approaches on new tabletop manipulation tasks that require diverse generalization. Finally, we deploy NOD-TAMP on a number of real-world tasks, including tool-use and high-precision insertion. For more details, please visit https://nodtamp.github.io/.

**Keywords:** Robot Learning, Robot Planning, Manipulation

## 1 Introduction

From children playing with Lego blocks to adults rearranging a room, our remarkable ability to plan long sequences of actions to achieve our goals is still beyond the capabilities of current robots. Consider the challenges involved in daily tabletop tasks shown in Fig. 1. First, these tasks are often *long-horizon* and full of sequential dependencies. Here, the robot must reason about the best pose to grasp a mug in order to stow it in a cabinet along with other steps to organize the entire table. Second, steps such as placing the mug in a tight cabin or stowing the screwdriver on the tool rack require *intentional contact*, which can render most motion planners that focus on avoiding collisions ineffective [1]. Finally, to be effective across broad environments, the robot must handle a wide *variation of object shapes* and *scene layouts*.

Task and Motion Planning (TAMP) [2, 3] is an effective approach for such problems because it can effectively resolve sequential dependencies through hybrid symbolic-continuous reasoning. However, TAMP systems typically require accurate, special-purpose perception systems and hand-engineered manipulation skills. Thus, it is difficult to apply them to unseen objects and tasks that require complex motion trajectories. Recent works have proposed to learn manipulation skills from demonstration [4, 5] to partially relax these constraints. However, their generalization ability remains bounded by the training data, which is costly to collect at scale [6].

By contrast, neural representation models have shown remarkable potential in enabling generalizable manipulation systems [7, 8, 9, 10]. In particular, Neural Object Descriptors (NODs) [8, 11, 12] are a powerful tool to extract dense, part-level features that generalize across object instances. Neural Descriptor Fields (NDFs) [8], a type of NOD that encodes SE(3) poses relative to a given object, can

---

*Equal Contribution.

8th Conference on Robot Learning (CoRL 2024), Munich, Germany.

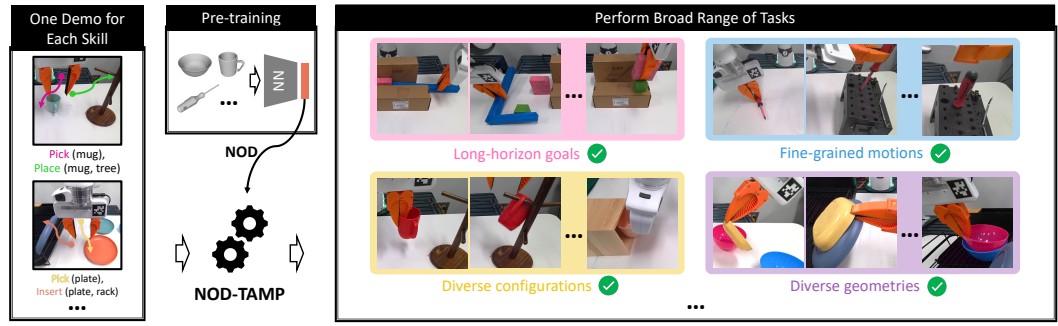

Figure 1: **Overview.** NOD-TAMP is a TAMP-based framework that adapts demonstration trajectories to new situations to accomplish long-horizon, fine-grained tasks.

adapt key-frame poses (e.g. grasps) for one object instance to others in the same object category (e.g. mugs), thereby achieving category-level generalization. However, existing NOD-based methods [8, 13, 14] are limited to adapting individual key-frame poses and thus struggle tasks involving complex motion and multi-step reasoning.

In this paper, we propose to combine these complementary paradigms and introduce NOD-TAMP, a TAMP-based framework that extracts adaptable skills from a handful of human demonstrations using NOD features and composes them to solve long-horizon tasks. Central to NOD-TAMP is a skill reasoning module that composes short-horizon skills to solve novel long-horizon goals that were never demonstrated, thereby achieving *compositional generalization*. To synthesize fine-grained manipulation trajectories for new objects, we propose a NOD-based trajectory adaptation module that can consistently adapt a recorded skill trajectory according to the observed objects. Finally, NOD-TAMP flexibly integrates the adaptation of recorded trajectories with traditional motion planning to generalize across drastically different scene layouts.

We empirically evaluate NOD-TAMP on many simulated multi-step manipulation tasks that test different factors of generalization across long-horizon tasks, including object shapes, number of objects, scene layout, task length, and task objectives. We find that NOD-TAMP can solve existing manipulation benchmarks [15], with a fraction (4 vs. 500 demos) of the data required by behavioral cloning methods. On a new task suite that stress-test generalization capabilities, NOD-TAMP also outperforms other existing methods [8, 16], some of which share a subset of its traits, highlighting the value of building a cohesive manipulation planning system. Finally, we successfully demonstrate NOD-TAMP on 6 real-world manipulation tasks.

## 2   Related Work

**TAMP.** Task and Motion Planning (TAMP) is a powerful paradigm for addressing long-horizon manipulation challenges by decomposing a complex planning problem into a series of simpler sub-problems [2, 17, 18, 19, 3]. Nonetheless, TAMP techniques presuppose knowledge of the object models and the underlying system dynamics. Such presuppositions can be limiting, particularly for domains with diverse objects and complex physical processes such as contact-rich manipulation.

**Learning for TAMP.** Recent works have set to address such limitations by replacing hand-crafted components in a TAMP system with learned ones. Examples include environment models [20, 21, 22, 23, 24], object relationships [25, 26, 27], skill operator models [28, 4] skill samplers [29, 30], and learned policies [31, 32, 33]. However, these learned components are often limited to the tasks and environments that they are trained on. Two notable exceptions are M0M [34] and GenTP [35], but both methods plan with predefined manipulation skills. In contrast, our work directly tackles the generalization challenge at the level of motion generation. Closely related to our work are methods that learn manipulation skills for TAMP systems [4, 36, 37]. However, the resulting systems remain bottlenecked by the generalizability of the skills, which are trained using conventional Reinforcement Learning [36] or Behavior Cloning [4, 37]. Instead, our work develops TAMP-compatible skills with object category-level generalization.

**Learning from Human Demonstrations.** Modern deep imitation learning techniques have shown remarkable performance in solving real-world manipulation tasks [38, 39, 40, 6, 41, 42]. However, the prominent data-centric view of imitation learning [43, 6, 42], i.e. scaling up robot learning via brute-force data collection, remains limited by the sample efficiency of the existing learning algorithms and the challenges in collecting demonstrations for long-horizon tasks in diverse settings. Other recent works have proposed to replay a small set of human demos in new situations to facilitate sample-efficient generalization [16, 44, 45, 46, 47, 48, 49, 50], but replay without adaptation can fail for novel object instances. Some other works leverage pretrained object representations to dramatically improve the generalization of policies given a handful of demonstrations [10, 8, 14]. However, these methods are limited to adapting a short skill [10] or a single manipulation action [8]. Our work develops a long-horizon planning framework that seamlessly integrates skills augmented with latent object representations into a classical TAMP framework.

## 3 Problem Setup and Background

The central question we aim to answer is: *given a set of demonstration trajectories, can we adapt and recompose segments of them to solve new tasks?* Our solution adopts the TAMP framework, where a high-level planner orchestrates a set of short-horizon motion generators (skills) to produce coherent long-horizon plans. The framework allows us to divide the problem into three technical sub-problems. (1) How to represent demonstration trajectory snippets as TAMP skills? In particular, how should we represent their precondition and effect constraints? (2) How to adapt skills instantiated with recorded trajectories to new scenes and objects? (3) Given a new task goal, how to chain these skills together to generate a trajectory plan? Our insight is that NOD features will enable us to adapt both motion trajectories and skill constraints to new scene layouts and object shapes. Our goal is to develop a cohesive TAMP framework that addresses these sub-problems by building its core components on NOD representations.

### 3.1 Problem Setup

We consider the problem of object rearrangement, where a robot must manipulate objects to achieve a desired scene configuration. The robot observes the scene in RGB-D frames and uses off-the-shelf segmentation models [51] to extract instance point cloud $P_o \in \mathbb{R}^{N \times 3}$ for each manipulable object $o$. Accordingly, we represent the environment state as a set of object point clouds and the robot end-effector pose $s = \langle \{P_o\}, T_w^e \rangle$, where $T_w^e \in \text{SE}(3)$ is the end-effector pose in the world frame. The goal is specified as a set of task-relevant object point clouds $g = \{P_o\}$, for example, a mug inside a cabinet. The robot must generate a sequence of actions $[a_1, ..., a_T]$ that manipulate the objects to reach a final configuration that closely matches the goal $g$, each action is an end-effector pose in the world frame $T_w^e \in \text{SE}(3)$. We measure task success by checking whether the desired scene configuration is reached. Our framework assumes access to a set of demonstration trajectories $\{\tau_i\}$, each of which is a sequence of actions $\tau_i = [a_0^{(i)}, a_1^{(i)}, ..., a_T^{(i)}]$, and the object point clouds capturing the initial state of the recorded scene. The objective is to adapt and compose the trajectories to generate action plans for solving a new task given a new scene layout with unseen objects.

**Neural Descriptor Fields (NDF).** Our approach leverages Neural Descriptor Fields (NDFs) [8] to compactly represent object poses and features. An NDF is a learned function $\psi_{\text{NDF}}$ that maps an object point cloud $P \in \mathbb{R}^{N \times 3}$ and a query pose $T^q \in \text{SE}(3)$ in the same frame to a feature descriptor $z \in \mathbb{R}^d$: $z \leftarrow \psi_{\text{NDF}}(T^q \mid P) \in \mathbb{R}^d$ (1). We focus on two key properties of NDFs: *Intra-category consistency*: For objects of the same category (e.g., mugs), a trained $\psi_{\text{NDF}}$ maps geometrically similar query points (e.g., mug rims) to similar feature descriptors $z$. *Pose invariance*: The descriptors are invariant to the object's global pose $T_w^o$, enabling generalization to new layouts.

We use NDF to solve an inverse problem: given a query pose $T_w^q$ and its feature $z$ derived from object point cloud $P_o$, recover the pose $T_w^{q'}$ relative to a new object cloud $P_{o'}$. This optimization problem can be solved with gradient descent: $\text{NDF-OPTIMIZE}(P_{o'}, z) \equiv \underset{T_w^{q'}}{\text{argmin}} ||z - \psi_{\text{NDF}}(T_w^{q'} \mid P_{o'})||$ (2).

### 3.2 Skill Representation

We employ NDFs in our skills to represent not only their control trajectories but also their start and end states. Accordingly, we represent each skill $\pi$ as a tuple: $\pi = \langle name, param, pre, eff, traj \rangle$.

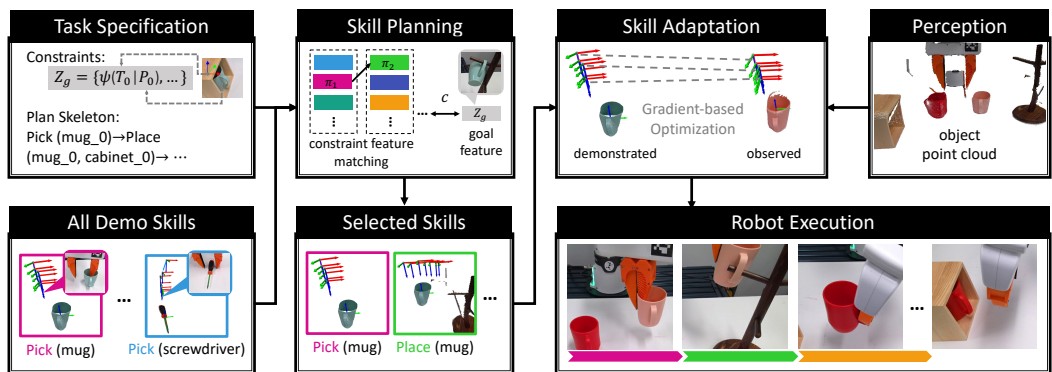

Figure 2: **NOD-TAMP Pipeline.** Given a goal specification, a task planner plans a sequence of skill types. Then, a skill reasoner searches for the combination of skill demonstrations that maximizes compatibility. Using learned neural object descriptors (e.g., NDFs), each selected skill demonstration is adapted to the current scene. Finally, the adapted skills are executed in sequence.

We show a concrete example of skill representation in supplementary material Fig. 16. Here, *name* denotes the skill type (e.g., PICK, INSERT). $param \triangleq \{o : P_o \in \mathbb{R}^{N \times 3} | o \in \mathcal{O}_\pi\}$ are skill parameters, which include the skill-relevant objects $\mathcal{O}_\pi$ and their observed point clouds. Preconditions *pre* specifies a set of constraints that must hold before skill execution. The *eff* is change in conditions, i.e., adding or removing constraints, as a result of successfully executing the skill. A *constraint* is represented by the relative configuration of two point clouds, which is encoded through NDF features (e.g., $z_{pre} \in \mathbb{R}^d$). For example, an INSERT skill may require the robot to hold a object in a specific way. Executing the skill results in a new constraint between the object and a receptacle. Finally, let $traj = \tau_i$ be a set of end-effector poses for this skill. A core objective of our method is to compose coherent multi-step plans by selecting and adapting a suitable trajectory within each constituent skill. We include details of all skills in our experiments in the supplementary material.

## 4 NOD-TAMP

We present NOD-TAMP, a method for adapting and recomposing a set of skill demonstrations to solve new tasks. First, we show how a single skill can be adapted to a new environment using NDFs (Sec. 4.1). Then, we propose a planning algorithm that identifies skill segments from multiple demonstrations to maximize compatibility (Sec. 4.2). Finally, we use motion planning to connect each skill in order to efficiently and robustly generalize to new environments (Sec. 4.3). The workflow for NOD-TAMP is illustrated in Fig. 2.

### 4.1 Skill Adaptation

We seek to adapt a skill to a newly observed scene, which may be populated with new objects and layouts. To do so, we leverage a key invariance: the skill trajectory still needs to satisfy the recorded constraints (i.e., relative configurations between pairs of objects). Our skill adaptation module (1) transforms the skill trajectories $\tau$ to *constraint-centric* NDF feature trajectories $\mathcal{Z}_\tau = [z_1, z_2, ...,]$ using Equation 1 and (2) adapts the trajectory to the observed scene via sequential optimization using Equation 2. Specifically, to encode a recorded skill trajectory and adapt it to the test scenarios, we consider common rearrangement skills that can be divided into two categories: hand-object interaction, such as grasping and manipulating constrained mechanisms (doors, etc.), and object-object interaction, where a robot uses the object in hand to interact with another object, such as placing and insertion.

**Hand-object interaction.** In this case, the constraint is between the manipulated object $o$ and robot end-effector $e$ at the end of a trajectory. Thus we use the recorded object point cloud as the conditioning input to the NDF to encode the demonstrated robot end-effector trajectory as an NDF feature trajectory $\mathcal{Z}_\tau = [z_1, z_2, ...,]$ where $z_i = \psi_{\text{NDF}}(T_w^e[i] \mid P_o)$.

**Object-object interaction.** Here, the constraint is between a pair of objects $(o, o')$, where $o'$ is the in-hand *source object*, and $o$ is a *target object*, e.g., a receptacle. Assuming a rigid transform

between the end-effector and $o'$ (i.e. a secure grasp), we can place a query pose $T_w^{q'}$ on $o'$ and create an object-centric demonstration trajectory. The encoded trajectory is thus $\mathcal{Z}_\tau = [z_1, z_2, ..., ]$, where $z_i = \psi_{\text{NDF}}(T_w^{q'}[i] \mid P_o)$. Note that the robot may also manipulate an unseen in-hand object $o'$ during deployment. For example, the robot may be asked to stow a larger mug in a bin when the demonstration is with a small mug. Thus we must also featurize the query pose with respect to $o'$ in order to satisfy the precondition constraint between $o'$ and the end-effector. To do so, we encode the constraint between the query frame and object $o'$ as $z_q = \psi_{\text{NDF}}(T_w^{q'} \mid P_{o'})$. This way, with feature $z_q$ and $\mathcal{Z}_\tau$, we can fully characterize the constraints between object $o$ and $o'$ across time while considering their shapes.

**Trajectory adaptation.** Given the feature trajectory $\mathcal{Z}_\tau$, we will use the NDF function conditioned on the observed point cloud to recover a transformed skill trajectory, i.e., $\text{NDF-OPTIMIZE}(P_{\text{observed}}, z)$, for each $z \in \mathcal{Z}_\tau$. We employ a sequential optimization procedure to speed up the convergence, where each optimized pose serves as the initialization to warm-start the optimization of the next pose. In the case of object-hand interaction, the optimization output is an end-effector trajectory that can be directly used as a sequence of robot control setpoints. In the case of object-object interaction, the output is an object-centric trajectory (the constraint between query frame and the receptacle object across time), which we need to convert to robot controls. To do this, we first adapt the constraint between the query frame and the in-hand object in test scenarios using the recorded feature $z_q$, resulting a rigid transform between the query frame and the in-hand object. Then with the in-hand pose, we can derive the end-effector poses for control. We describe the skill adaptation with extended notation and pseudocode algorithm in the supplementary material.

## 4.2 Skill Planning

Given a set of skills, the goal of the skill planner is to select a sequence of skills and their motion trajectories that can be chained together to reach a task goal. The trajectories are then adapted using the procedure described in Sec. 4.1.

While our system can generate task plans independently or be embedded in an outer TAMP algorithm thanks to our skills functioning as PDDL operators, we assume the task plan is given. Our focus is on the bi-level planning problem of chaining and adapting skill trajectories to create coherent long-horizon motion plans. For a given task, we assume an $H$-step task plan skeleton $[\hat{\pi}_1, ..., \hat{\pi}_H]$ that defines a sequence of selected skills, e.g., [PICK(mug), PLACE(mug, bin), ...]. Recall that each skill can contain multiple candidate demonstration trajectories. The start and end of each trajectory represent its precondition and effect constraint, respectively. The essential step in skill planning is, for each skill in the plan, choose a candidate trajectory that is most compatible with the constraints of its adjacent skills. We calculate compatibility based on the distance between pairs of constraints in the NDF space. For simplicity, for the $i$-th skill in the plan, we denote the NDF-encoded precondition of a candidate skill trajectory as $z_{\text{pre}}^i$, and the effect as $z_{\text{eff}}^i$. The compatibility is calculated as $c = ||z_{\text{pre}}^i - z_{\text{eff}}^{i-1}|| + ||z_{\text{eff}}^i - z_{\text{pre}}^{i+1}||$. Finally, we parse the goal configuration $g$ into a set of pair-wise object constraints and encode them as a set of NDF features $Z_g$. We then compute the plan cost as the distance between $Z_g$ and the final accumulated constraints of the entire plan sequence.

We use Depth-first search (DFS) to traverse over possible skill plans. After we obtain all costs for all skill trajectory combinations, the plan with the lowest plan-wide NDF feature distance is returned. For simplicity, we present this as a Cartesian product over relevant skills, but this can be done more efficiently by performing a Uniform Cost Search in plan space, where the NDF feature distance serves as a the cost function. Algorithm 2 in the supplementary material displays the pseudocode for the NOD-TAMP planner.

## 4.3 Transit & Transfer Motion

Adapting demonstrated skills is particularly effective at generating behavior that involves contact. However, demonstrations typically contain long segments without contact (outside of holding an object). Because these components do not modify the world, it is often not productive to replicate them. Thus, we temporally trim skill demonstrations to focus on the data points that involve contact. In our implementation, we simply select the 20 steps before contact.

After trimming, and in often before trimming, two adjacent skills might be far away in task space. While linear interpolation is an option, this is not generally safe because the straight-line path may cause the robot to unexpectedly collide. To address this, we use motion planning to optimize for safe and efficient motion that reaches the start of next the skill. Motion planning generally requires some characterization of the collision volume of the obstacles to avoid. Because we do not assume access to object models, we use the segmented point clouds as the collision representation. For each pose yielded by the skill, we use Operational Space Controller (OSC) [52] to track them.

## 5 Experiments

We validate NOD-TAMP and how its components contribute to solve long-horizon tasks, perform fine-grained manipulation, and generalize to new object shapes. We select three evaluation settings: (1) LIBERO [15], a standard manipulation benchmark that feature diverse objects and long-horizon tasks, (2) a set of custom tabletop tasks that stress test spatial generalization and skill reasoning&reuse, and (3) six real-world tasks with noisy perception and multitudes of challenges combined. We highlight key conclusions here and leave additional details in the supplementary material.

### 5.1 Experimental Setup

**LIBERO Benchmark:** LIBERO [15] is an existing multi-task manipulation benchmark. Our evaluation covers the "LIBERO-Spatial" (10 tasks), "LIBERO-Object" (10 tasks), and three of the "LIBERO-Long" tasks (Task 1, 5, and 8). For "LIBERO-Spatial" tasks, we provide our system with just one demo of manipulating a bowl instance and test each system's ability to generalize over different initial bowl poses and goal configurations. For "LIBERO-Object" tasks, we provide our system with four demos of manipulating a cheese box, milk box, ketchup bottle, and soup can and then test our system's ability to generalize over similar objects shapes (e.g., salad dressing bottle, pudding box) and poses.

**Customized Tabletop Tasks:** To push the limit of the system, we design a suite of rearrangement tasks that have large variation in task horizon, object instances, scene layouts, goal configurations, and precision tolerances (See Fig 3)."MugPicking" - Pick up mugs with varying shapes and initial poses; "MugInsertion" - Insert mugs of varying shape into a tight cabinet. Both the mug and the cabinet are randomly placed on the table; "TableClear" - Place two mugs into two bins, which aims to test the ability to achieve long-term goals by reusing the skills; "TableClearHard" - Stow one mug into a cabinet with side opening and place another mug into a bin. The robot must reason about proper grasp strategy to achieve the goal; "ToolHang" [39] - Insert the frame into a stand with tight tolerance, and then hang the tool object on the inserted frame, which tests the cabability of handling fine-grained motions.

We provide only **two** demos, which manipulate **one** mug instance in two different ways by grasping either the handle or the rim, and test the methods on other **nine** different mug shapes. For the "ToolHang" task, we provide **one** demo of how to insert the frame and hang the tool on the frame after it is assembled.

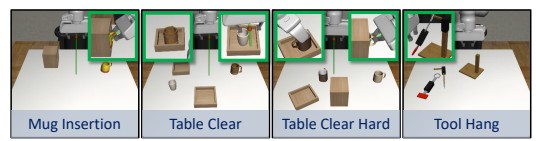

Figure 3: **Customized tasks.** Examples of initial state and goal state (in green bounding box).

### 5.2 Baselines

**NDF+** [8] - We augment the original NDF method, which has only shown single-pose optimization, with task skeleton and the skill planning module. This baseline also uses a motion planner to transition between key-frame poses; **MimicGen+** [16] - MimicGen directly transforms the demonstrated poses to the relevant object frame and then sent to the controller without further adaptation. For fair comparison, we augment MimicGen with a motion planner for collision avoidance; **BC** - The best-performing BC baseline (ViT-T) from LIBERO benchmark [15]. We list the reported performance in the multi-task learning setting as it is an upper bound for lifelong imitation learning.

We also compare our full system with different variants: **Ours/SR** - This ablation removes the skill planning module. For each skill, we randomly choose a reference trajectory from the collected

demonstrations belonging to this skill. This baseline validates the importance of skill reasoning for generalizing across tasks; **Ours/MP** - This ablation removes the motion planning component and uses linear trajectory interpolation to achieve transitions between the adapted skill trajectories. This baseline validates the benefit of leveraging motion planning, a capability present in TAMP systems; **NSC** (naïve skill chaining) - This baseline ablates both the skill reasoning and the motion planning component, it randomly selects a reference trajectory for each skill, adapts the skill with NDF, and uses linear trajectory interpolation for transitions between the selected trajectories.

## 5.3 Evaluation on the LIBERO Benchmark

This experiment compares the best behavior cloning (BC) performance provided by the LIBERO benchmark [15] with methods that combine generalizable neural representations and model-based planners, such as NOD-TAMP and several baselines (See Tab. 1). For the "`LIBERO-Spatial`" and "`LIBERO-Object`" tasks, the BC method is trained with 500 demos, and achieves 78% success rate. In contrast, our system only requires namely 1 demo for "`LIBERO-Spatial`" and 4 demos for "`LIBERO-Object`", the success rates are 84% and 94% respectively.

Table 1: Success rates on LIBERO tasks. MimicGen$^+$, Ours/MP, and Ours/SR are abbreviated as M$^+$, O/MP, and O/SR.

| Tasks | BC | NDF$^+$ | MG$^+$ | NSC | O/MP | O/SR | Ours |
|---|---|---|---|---|---|---|---|
| Spatial | 0.78 | 0.72 | 0.82 | 0.74 | 0.72 | **0.86** | 0.84 |
| Object | 0.78 | 0.80 | 0.88 | 0.80 | 0.76 | 0.90 | **0.94** |
| Long1 | 0.80 | 0.50 | 0.40 | 0.10 | 0.10 | **0.70** | 0.70 |
| Long5 | 0.52 | **0.70** | 0.60 | 0.20 | 0.20 | 0.60 | **0.70** |
| Long8 | 0.00 | 0.30 | 0.80 | 0.20 | 0.10 | 0.80 | **0.90** |

We hypothesize that the performance gap is caused by different state/action representations and the structural information leveraged to build the system. The BC methods directly learn a mapping from the scene observation to the actions and thus require a huge broad data to cover diverse situations. Our method, along with the action-transferring baselines (e.g., NDF, MimicGen), utilize object-centric representations to adapt the spatial correspondences from demo scenes to test scenes. MimicGen$^+$ assumes identical correspondences between demonstration objects and test objects, directly replaying trajectories in the local frames of test objects. In contrast, our approach leverages learned object representations to infer spatial correspondences, enabling the transferred actions to be more robust to variations in object geometry. Compared to NDF$^+$, NOD-TAMP transfers a sequence of actions that represent each dynamic manipulation skill instead of just a single target state, in this case, the last end-effector pose in the demonstration trajectory. This improves NOD-TAMP's performance in fine-grained manipulation tasks. As LIBERO tasks are less constrained on grasping strategies (e.g., grasping the bowl to place it on a plate), we found the O/SR baseline performs similarly to our full system.

## 5.4 Evaluation on Customized Tabletop Tasks

In the tabletop tasks, NOD-TAMP consistently achieves a high success rate (80-90%) across all tasks and outperforms the other baselines and ablations (see Tab. 2). Below, we highlight specific comparisons and underscore the importance of each component in NOD-TAMP. Additional analysis is in supplementary material.

**NOD-TAMP exhibits strong performance across long-horizon tasks and is able to reuse skills in new contexts.** The "`TableClear`" task requires re-using the existing **two** pick-and-place human demonstrations, which only consisted of single mug and bin interactions, to stow two mugs into two bins. NOD-TAMP

Table 2: Success rates on customized tabletop tasks. MimicGen$^+$, Ours/MP, and Ours/SR are abbreviated as M$^+$, O/MP, and O/SR.

| Tasks | NDF$^+$ | MG$^+$ | NSC | O/MP | O/SR | Ours |
|---|---|---|---|---|---|---|
| MugPicking | 0.80 | 0.70 | **0.85** | 0.80 | **0.85** | **0.85** |
| MugInsertion | 0.75 | 0.55 | 0.80 | 0.85 | 0.80 | **0.90** |
| TableClear | 0.60 | 0.75 | 0.80 | 0.75 | **0.85** | 0.85 |
| TableClearHard | 0.40 | 0.55 | 0.15 | 0.50 | 0.10 | **0.80** |
| ToolHang | 0.00 | 0.35 | **0.75** | 0.70 | **0.75** | 0.75 |

achieves strong performance and outperforms MimicGen$^+$ by 15% and NDF$^+$ by 25% on this task, showcasing a superior ability on re-purposing short-horizon skills for long-horizon manipulation.

**NOD-TAMP exhibits strong generalization capability across goals, objects, and scenes in long-horizon tasks.** The "`TableClearHard`" task requires intelligent selection and application of demonstration trajectories to achieve diverse mug placements. We see the clear benefit of the skill planning component to achieve the different goals in this task – NOD-TAMP outperforms Ours/SR by 70% and NSC by 65%. The omission of the skill planning module results in an incompatible

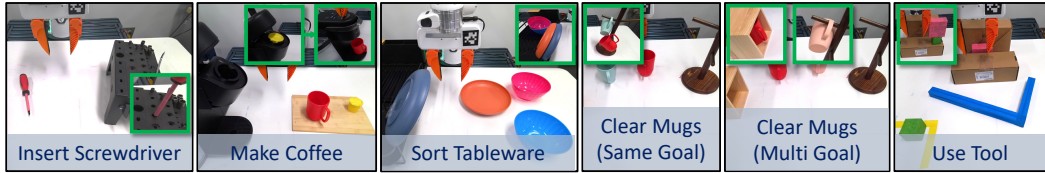

Figure 4: **Real-world tasks.** Examples of initial and intermediate / goal states.

composition of skills. For example, the robot may grip the rim of a mug and attempt to insert it into the cabinet, leading to collisions between the cabinet and the gripper.

**NOD-TAMP is able to solve low-tolerance manipulation tasks.** The ToolHang task requires fine-grained manipulation skills that adapt to various object poses. NOD-TAMP achieves 75% success rate, outperforming NDF$^+$ and MimicGen$^+$ that cannot adapt their trajectories based on environment changes. Since this task does not require obstacle avoidance or constrained grasping strategies, our ablated baselines (Ours/MP, Ours/SR, and NSC) perform similarly to our full method.

### 5.5 Real-world Evaluation

We deploy NOD-TAMP on a real Franka Panda robot to solve six challenging manipulation tasks (Fig. 4): "SortTableware" - Insert a dish into a narrow slot on the rack and stack two bowls on top of it; "MakeCoffee" - Place a mug under the coffee machine, insert a coffee pod into a tight holder, close the lid and then press the button; "InsertScrewdriver" - Insert a screwdriver into a tight slot on the storage rack; "UseTool" (inspired by [19, 53]) - Use the L-shape tool to poke a box out from a narrow tunnel, and hook another box that out of reach, and stack them; "ClearMugs(SameGoal)" - Hang two mugs on the mug tree; "ClearMugs(MultiGoal)" - Hang one mug on the tree, and insert another mug into the cabinet. We provide a single demonstration for each ⟨skill, object category⟩ pair and test skill reasoning and reuse across tasks, object instances (e.g., round vs. square plates), and scene configurations. We include more details, including the list of object instances and reset range in the supplementary material.

We use a front-mounted Microsoft Azure Kinect camera to capture RGB-D images and SAM [51] to segment object point cloud. NOD-TAMP plans directly based on the partial-view object point cloud and executes the plans with impedance control. NOD-TAMP achieves a 60% success rate on "MakeCoffee", 70% success rate on "InsertScrewdriver", showing its capability on handling fine-grained motions (e.g., inserting the coffee pod or screwdriver with tight tolerance). It achieves 90% success rate on "SortTableware" and "UseTool", and 80% success rate on "ClearMugs(SameGoal)" and "ClearMugs(MultiGoal)", suggesting its capability on skill reusing and reasoning based on long-horizon goals, e.g., how to grasp the mug to store into bin vs. hang on mug tree and grasping different part of the tool to poke and hook.

## 6 Conclusions and Limitations

We introduced NOD-TAMP, a planning algorithm for long-horizon and fine-grained manipulation that can generalize across object shapes. NOD-TAMP directly leverages human demonstrations to implement manipulation skills. To ensure that these skills generalize to new settings, NOD-TAMP uses NDFs to adapt demonstrated object-centric motion to new, unseen objects. These skills are chained together using feature matching to ensure plan feasibility. Finally, they are executed using traditional motion planning and control to generalize across environments. We evaluated NOD-TAMP and competitive baselines on two simulated task suites and six real-world tasks.

A limitation of NOD-TAMP is the computational efficiency in NOD-based trajectory adaptation. This bottleneck might be addressed with lightweight neural networks or more efficient optimization techniques. For skill planning, we use DFS to traverse over possible skill plans, this part can be further accelerated through optimization in the implementation (e.g., parallel DFS or building K-D trees in the NDF constraint space) to accommodate large-scale demos. Besides, extending the system to handle deformable objects would necessitate adapting the current NOD to account for time-varying features, which represents a challenging but interesting area for future work.

# 7 Acknowledgment

This work was partially supported by NSF Award #2409016 and NSF Award #2101250.

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

# A Table of Contents

The supplementary material has the following contents:

- **Real-World Experiments** (Sec. B): Elaborate on our real-world experimental setting and analyze the results
- **Skill Reasoning Visualization** (Sec. C): Visualize the feature matching results for real-world trials to analyze the skill reasoning process
- **Simulation Experiments** (Sec. D): Elaborate on our customized tabletop task setting and analyze the results
- **Robustness to Perception Noise** (Sec. E): Perform experiments that analyze the effect of sensor noise on NOD-TAMP success rates
- **Computation Efficiency** (Sec. F): Break down the runtime of the different components of NOD-TAMP
- **Demonstration Extraction and Skill Representation** (Sec. G): Elaborate on the demo extraction process and skill representation
- **Demo Quality Analysis** (Sec. H): How demonstration quality affects performance
- **NDF Training** (Sec. I): Describe how we trained our NDFs
- **LIBERO Qualitative Results and Failure Modes** (Sec. J): Review failure modes on the LIBERO tasks
- **Pseudocode** (Sec. K): Present algorithms for trajectory adaptation and skill planning

# B    Real-world Experiments

## B.1    System Setup

We demonstrate deploying our method on a real Franka Emika Panda robot in Fig. 5. The system perceives the scene with a Microsoft Azure Kinect camera and uses the Segment Anything Model (SAM) [51] to generate instance segmentation masks. To identify the target objects for each task, we extract visual features for each mask region using a CLIP model [54], and retrieve the target masks through the text descriptions of target objects. We project the pixels belonging to each target object into the robot base frame to generate point clouds. For a pixel with coordinate $(u, v)$ and depth $d$, the corresponding 3D location can be recovered by

$$p = R \cdot K^{-1} \cdot I + t,$$

where $I = (ud, vd, d)$, $[R|t]$ denotes the camera pose obtained through calibration, and $K$ denotes the camera intrinsic matrix.

The motion planning component is built on [55]. We execute trajectories using open-loop control and track them with a joint impedance controller [56] operating at a frequency of 20 Hz.

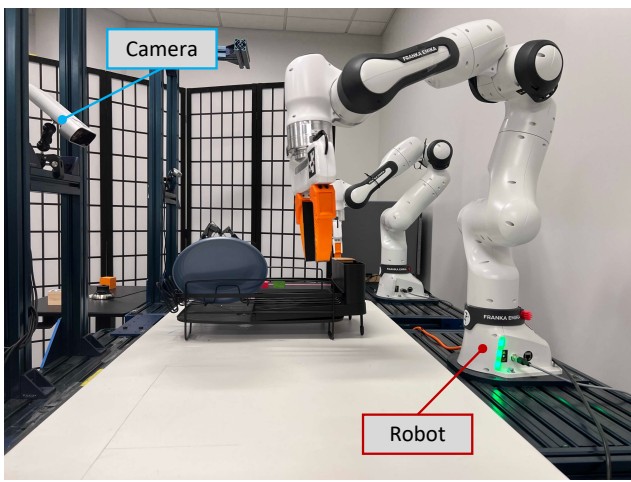

Figure 5: **Hardware Setup**. An illustration of the hardware setup.

## B.2    Task Details

The objects used in each task, an example of start/goal state, reset range, and skills recorded are illustrated in Fig. 6 and Fig. 7. The skills are extracted from a single demonstration of the full task. Below we describe each real world task setup and the skill demonstrated.

- "`InsertScrewdriver`": A fine-grained manipulation task. Pick up a screwdriver by the handle and insert it into a tight slot (approx. 5mm) on the storage rack.
- "`SortTableware`": A multi-step manipulation where the robot must place dishes onto a dish rack. Grasp and insert a dish into a narrow slot on the dish rack and stack two bowls next to it. Dishes and bowls vary in shapes and size in each evaluation trial.
- "`MakeCoffee`": Operate a Keurig machine to make coffee — a multi-step task with fine-grained manipulation steps. Pick up a mug by the handle and place it the coffee machine, insert a coffee pod into the tight holder, close the lid and then press the button.
- "`ClearMugs(SameGoal)`": Grasp and hang two mugs on the mug tree. The robot must reason about how to pick up the mug (by the rim, not the handle), in order to hang the mugs.
- "`ClearMugs(MultiGoal)`": Grasp and hang one mug on the mug tree, and grasp and stwo another mug into the cabinet. The robot must reason about how to pick up the mug: to hang the mug, pick up by the rim. To stow a mug, pick up by the handle.

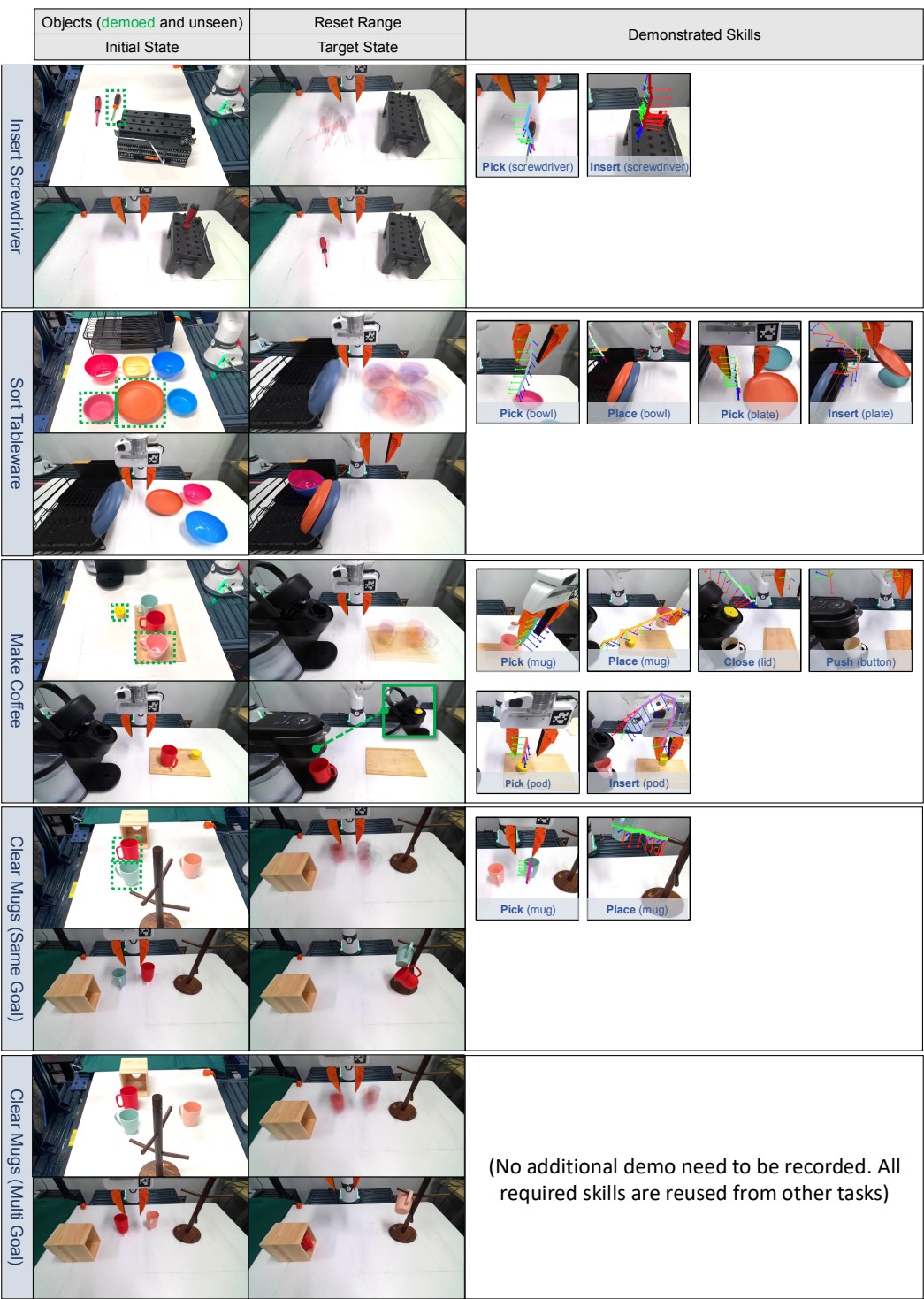

Figure 6: **Real-world Task Setup (part 1).** Visualization of the real world tasks. For each row, we show the objects used for the task. The objects used in the demonstration are visualized using bounding boxes with green dotted lines. We then show an example start and goal state, the reset range by overlaying the initial frames of each trial, and the skills recorded for the task.

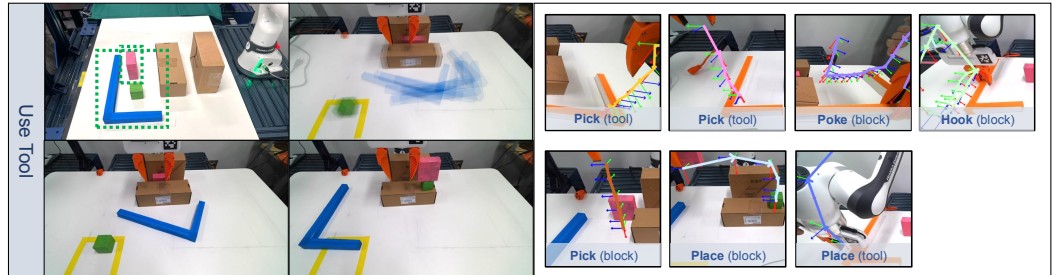

Figure 7: **Real-world Task Setup (part 2).**

- "UseTool": A classical physical problem-solving task that requires multi-step reasoning [19, 53]. The robot must reason about how to pick up the tool in order to use the tool to interact with objects in the scene. Grasp the junction of an L-shape tool and use it to poke a box out from a narrow tunnel, and then regrasp the long handle of this tool to hook another box that out of reach, and finally stack the two boxes.

**Skill reuse.** For the "ClearMugs(SameGoal)" task, we only record a demonstration of how to grasp a mug and hang it on the mug tree, as the skills of grasping a mug and stow it into cabinet can be re-used from the "MakeCoffee" task; For "ClearMugs(MultiGoal)", we do not record any new demonstration as all the required skills for this task can be re-used from the "ClearMugs(SameGoal)" task.

**Evaluation setup.** We conduct 10 evaluations per task. Select tasks involve different object instances for each evaluation. Objects are placed randomly within their respective initialization range.

## B.3 Performance Analysis

The quantitative results are shown in Tab. 3.

Table 3: Success rates of our system on real world tasks.

| Tasks | SortTableware | MakeCoffee | InsertScrewdriver |
|---|---|---|---|
| Success Rate | 9/10 | 6/10 | 7/10 |
| Tasks | UseTool | ClearMugs (SameGoal) | ClearMugs (MultiGoal) |
| Success Rate | 9/10 | 8/10 | 8/10 |

The task execution process is visualized in Fig. 8 and Fig. 9. NOD-TAMP can handle fine-grained motions (e.g., inserting the coffee pod or screwdriver with tight tolerance), and demonstrate its capability to re-use skills and reasoning over them to achieve long-horizon goals (e.g., grasping different parts of the tool to achieve poking and hooking behaviors). We also notice the major failures are caused by that the robot fails to grasp the handle of the mug, or not precisely align the pod with the holder of the machine, where the errors can be attributed to noisy depth perception, or incomplete object point clouds due to partial view observation. We conduct further analysis on perception noise in Sec. E.

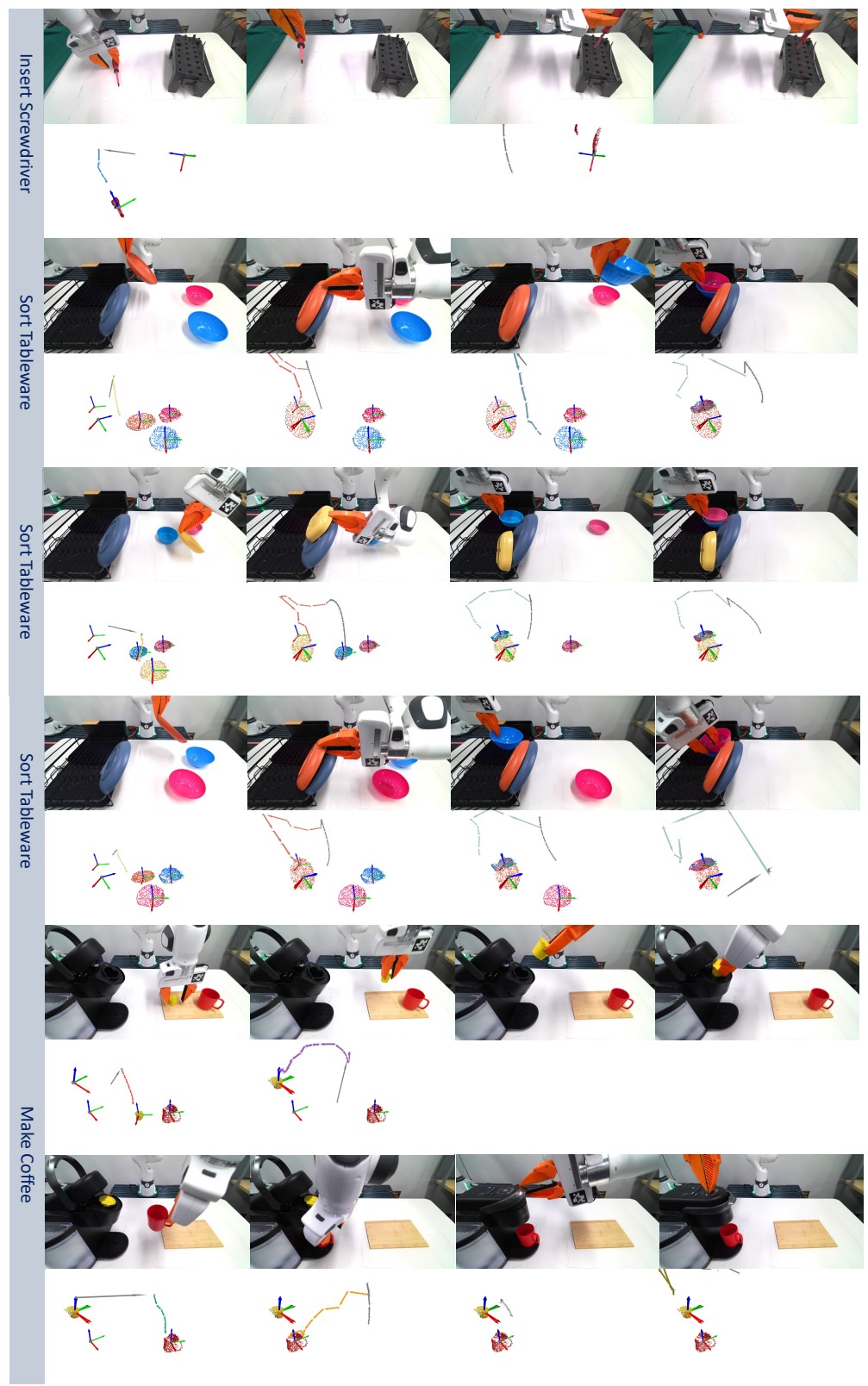

Figure 8: **Real-world Results.** Key frames of real world task execution processes, the planning results are shown below each frame.

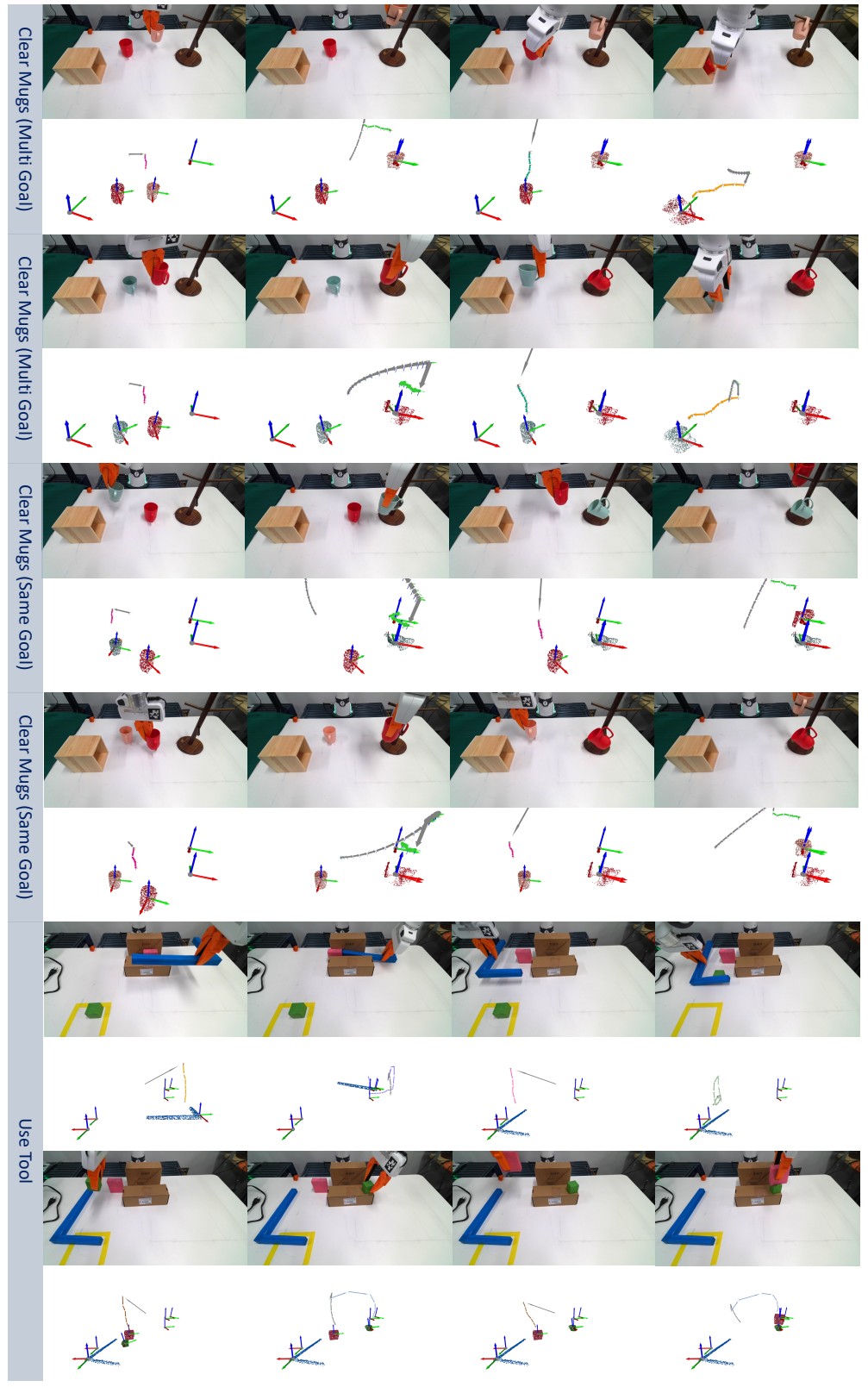

Figure 9: **Real-world Results (Continued).**

## C  Skill Reasoning Visualization

To analyze the skill planning process, we use real world trials of "`ClearMugs(SameGoal)`", "`ClearMugs(MultiGoal)`", and "`UseTool`" for visualizing the feature distance of different skill combinations (See Fig. 10). For the left part of the figure, each row represent different strategies of picking a mug (i.e., grasp the rim or grasp the handle), each column represent different strategies of placing the holding mug (i.e., insert the mug into cabinet, or hang it on mug tree). For the right part of the figure, each row represents different ways of pick up the tool (i.e., grasp either the junction or the long handle), and each column show different ways of using the tool (i.e., poke object out of a tunnel or hook object that out of reach). Lower feature distance means better compatibility of the skill combination. The results show that by leveraging the learned object descriptor features that characterizing geometric configurations, our skill planning module is able to correctly evaluate the skill compatibility.

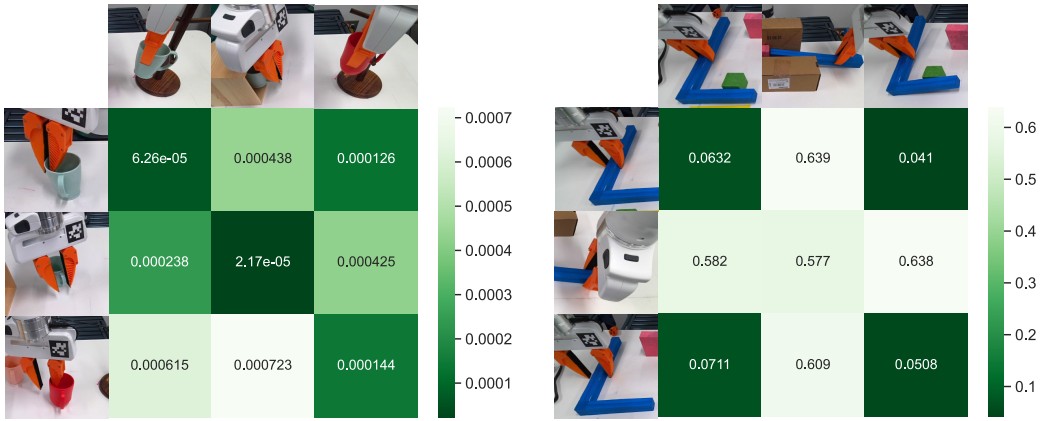

Figure 10: **Feature Matching.** The NOD feature distance of different skill combinations for real world trials. Lower score indicates more compatible skills (pre-post condition matching).

# D Simulation Experiments

We visualize all reference demos used in the customized tabletop tasks in Fig. 11. We record just one demo for each task and post process the recorded data into skills. We conduct 20 evaluation trials for each task, and we change the object shapes and poses for each trial to test the generalization of the system, we visualize the task reset ranges by overlying the first image frame of each trial in Fig. 12.

Fig. 13 show the generated trajectories of our framework and the execution process for our proposed tabletop tasks in simulation, highlighting the capability of our system on handling diverse shapes, configurations, and task goals.

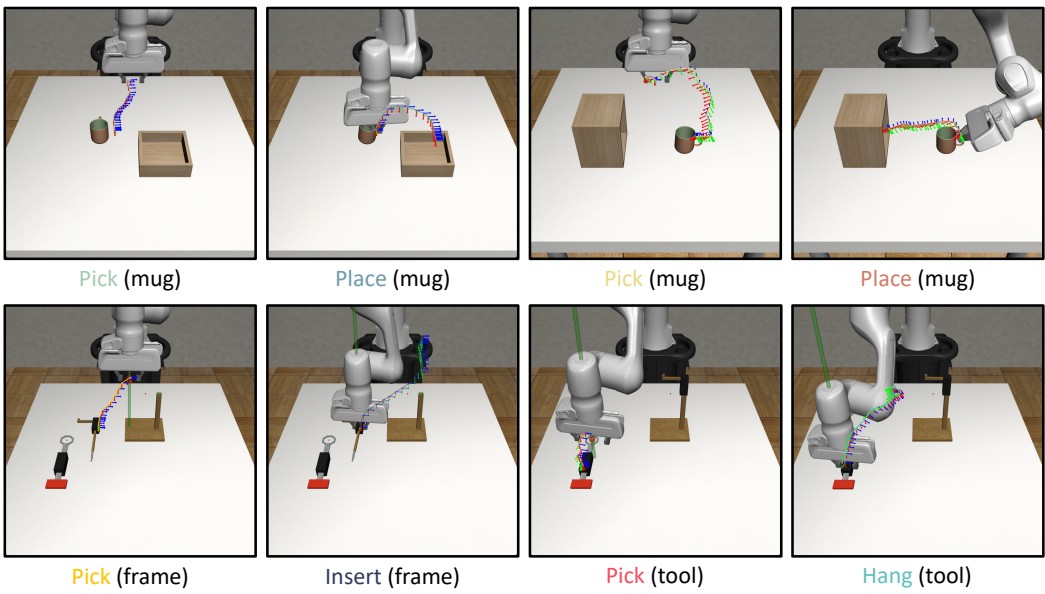

Figure 11: **Simulation Demos.** A visualization of the reference skill demos used for each customized tabletop task. Here, the trajectories for each skill are projected into the camera coordinate frame and drawn on top of the initial RGB image.

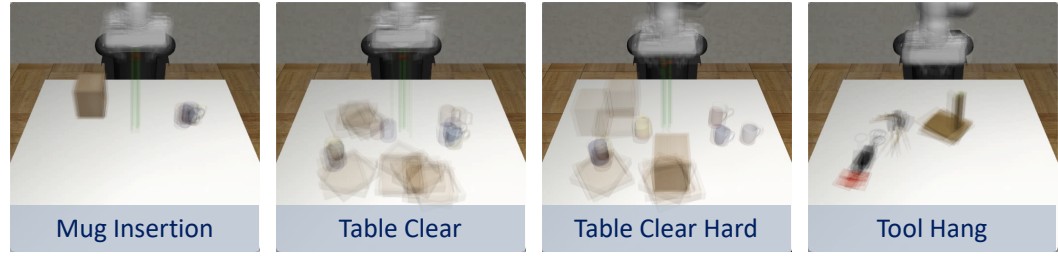

Figure 12: **Customized Tabletop Task Reset Ranges.** The task reset ranges.

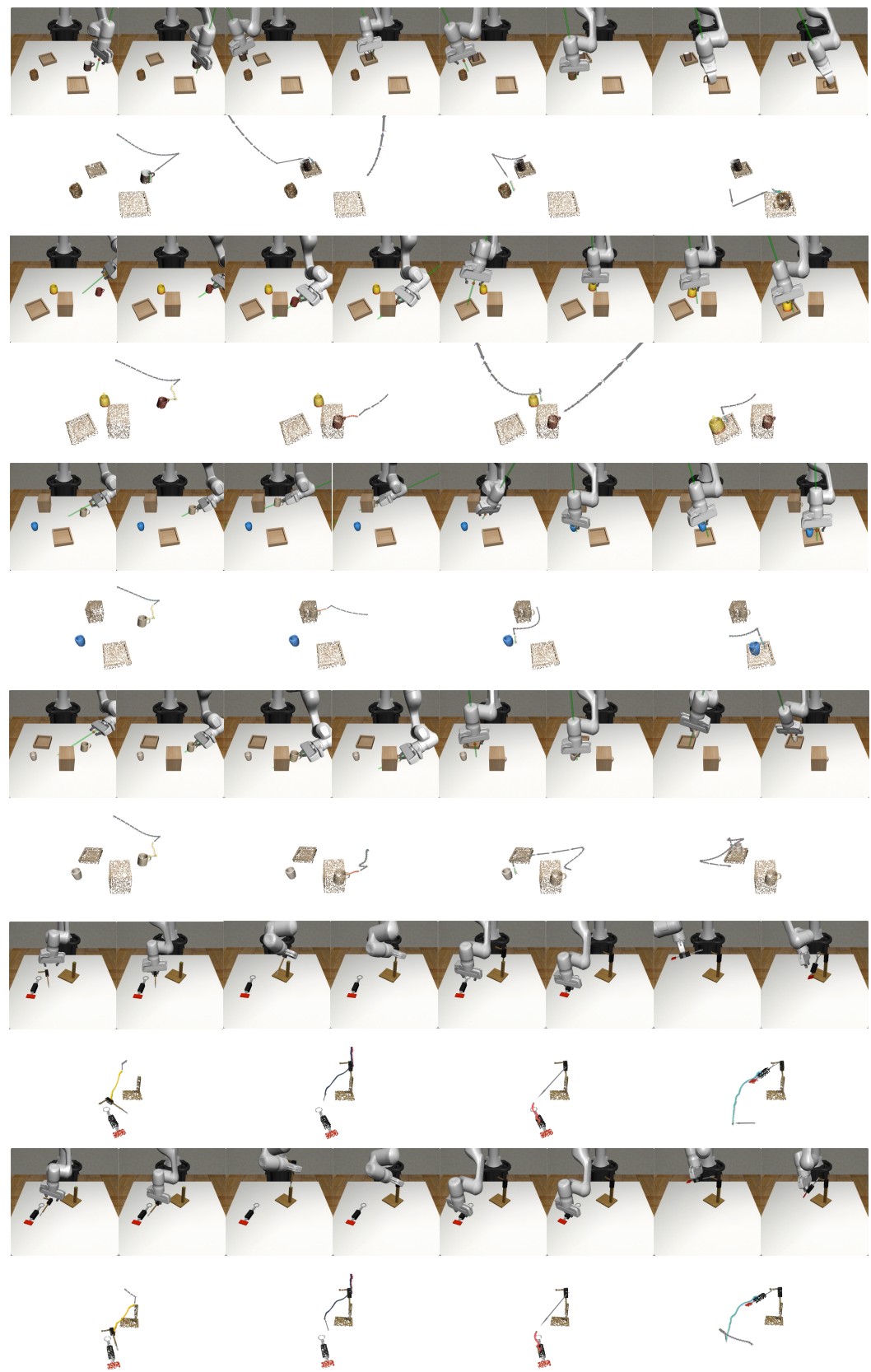

Figure 13: **Simulation Results.** Key frames of task execution and planning.

# E Robustness to Perception Noise

To understand the performance of our system under different levels of perception noise, such as levels present in real-world sensors, we perform an experiment where we inject noise in the point cloud observation in simulation. We peform evaluation on the first stage of the "`ToolHang`" task, a high-precision task with tolerance of approximately 5mm. The robot needs to pick up the frame object and insert it into a stand. A study of the depth accuracy of the Microsoft Azure Kinect [57] showed that, within a distance of $0.8$ meters, the noise standard deviation is $5.546 \times 10^{-4}$ meters.

To simulate this and settings with increased noise, we inject Gaussian noise with standard deviations of $0.05$, $0.1$, $0.15$, and $0.2$ centimeters. The results of the experiment are shown in Fig. 14. NOD-TAMP only experiences a 5% reduction in success rate for real-world levels of noise. Our experiments show that NOD-TAMP can robustly complete precise tasks even in the presence of typical sensor noise.

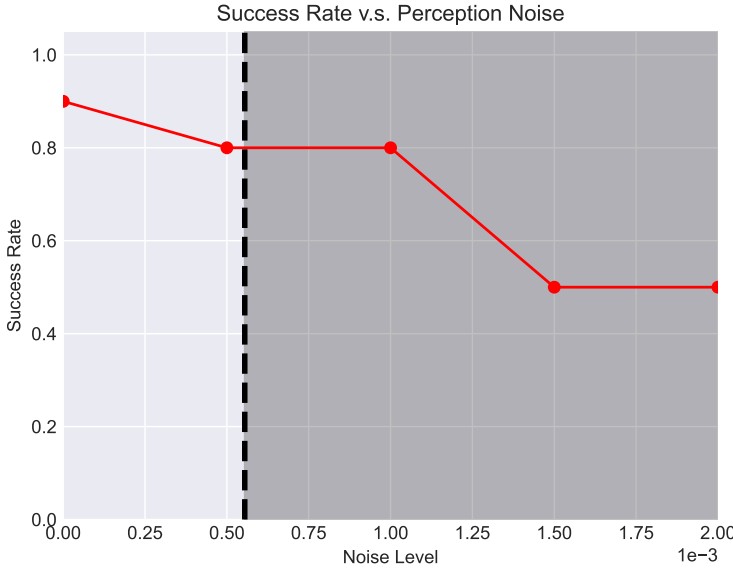

Figure 14: **Robustness to Perception Noise**. We evaluate the performance of NOD-TAMP under different levels of perception noise on the first stage of the simulated "`ToolHang`" task. The vertical dotted line represents the Gaussian noise standard deviation of a Microsoft Azure Kinect. The success rate of NOD-TAMP only slightly decreases for real-world levels of noise, indicating that NOD-TAMP is robust to sensor noise.

# F Computation Efficiency

We provide a planning runtime analysis of our system in Fig. 15. We evaluate NOD-TAMP on a two-stage task that involves skill chaining and reasoning. We report the runtime of the trajectory adaptation, constraint transfer & skill reasoning, and trajectory tracking components. Since most daily tasks can be achieved through sparsely represented trajectories with around 10-20 poses, altogether, the planning time is typically 1-2 minutes, where the gradient-based NDF optimization occupies most of the runtime.

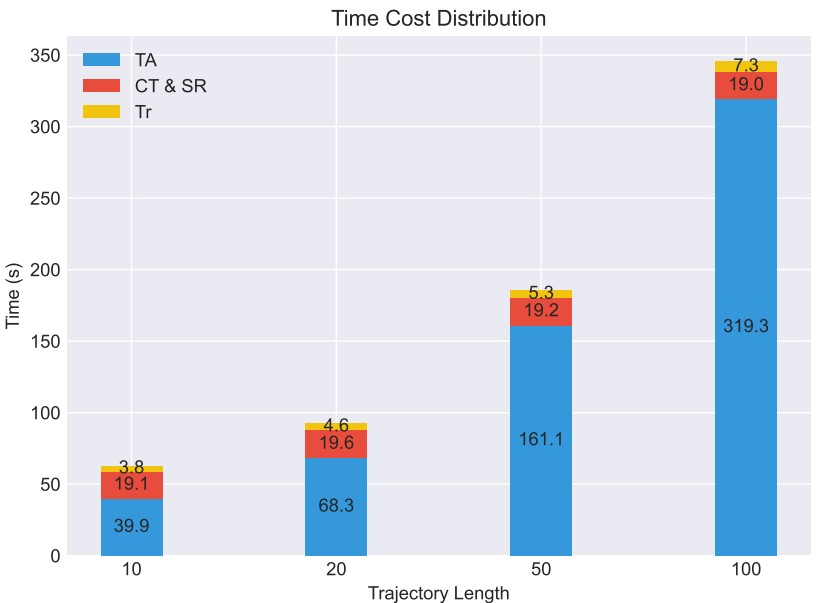

Figure 15: **Time v.s. Trajectory Lengths**. We show the runtime of our full system for a two-stage task. **TA** is short for trajectory adaptation, **CT & SR** is short for constraint transfer and skill reasoning. **Tr** is short for trajectory tracking. We see that trajectory adaption is the most computationally expensive operation in NOD-TAMP.

We also observe that the computational bottleneck is trajectory adaptation, which involves NDF optimization of individual poses to align with the reference trajectory feature. The runtime of this component can be improved by utilizing lightweight neural networks for feature encoding and leveraging more efficient optimization techniques. This is left for future work.

# G   Demonstration Extraction and Skill Representation

Here, we provide additional detail on segmenting and representing skills from recorded demos, to supplement Sec. III.B and IV.B-C in the main text. To segment skill-level demonstrations from a longer task demonstration, we identify kinematic switches, which can be detected from gripper open & close actions and contact. Specifically, we detect object contacts and pinpoint the time step at which these changes occur to establish the boundaries of each skill, similar to prior works that uses signals such as gripper-object contact [58]. Some data sources, such as LIBERO, contain noisy actions such as repeated grasps and accidental contacts. To correct for this, we manually inspect and filter out low-quality skill demonstrations. To better leverage transit and transfer motion planning, we trim the skill segments to be just the actions before changes in contact. In our implementation, we simply consider the 50 steps before contact. Further discussion is included in Sec. IV.C.

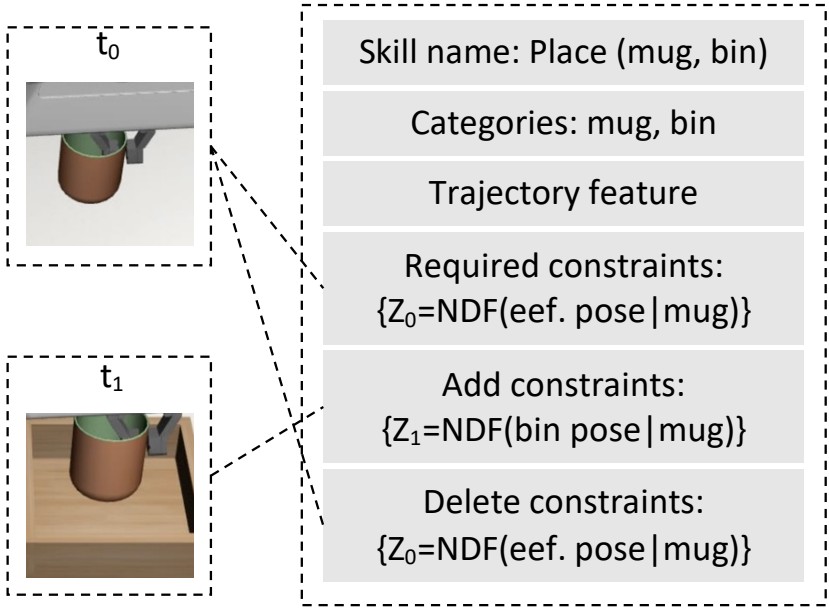

Figure 16: **Skill Representation**. How we represent a pick skill in NOD-TAMP: the "required constraints" represent preconditions and the "add & delete constraints" represent effects.

Fig. 16 illustrates how a skill is represented in the skill planning step (Sec. III.B). During skill planning, a candidate skill is currently executable only if the currently active set of constraints, which are updated after each skill is added to the current partial plan during the search, include the required constraints of the candidate skill. Additionally, we use a compatibility score in the form of the feature distance between two matched constraints to rank plan viability.

# H  Demo Quality Analysis

To analyze how demo quality affects the performance of our system, we use the "Can" task from Robomimic benchmark [59] to test our system, which paired with hybrid human demos. According to Robomimic, the demos are categorized into three groups with quality "better", "okay", and "worse". We randomly sampled 4 demos from each group, and we run 10 evaluation trials for each source demo with randomly initialized object placements, the results are presented in Fig. 17.

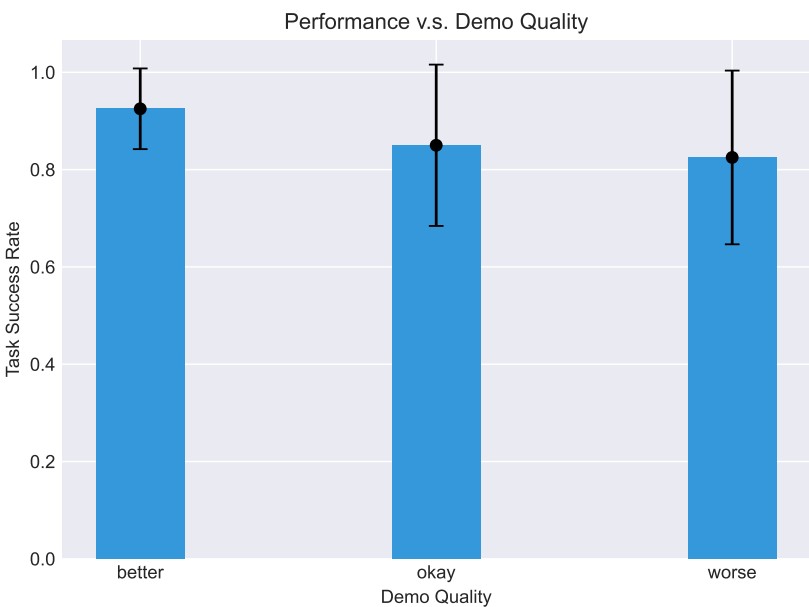

Figure 17: **Task success rate v.s. Demo Quality** for the "Can" task in Robomimic [59]. Demonstrations of different qualities are extracted from the accompanying dataset.

The results show that our system's performance is affected by the quality of the reference demonstrations, similar to other learning from demonstration methods. However, the performance degradation is minimal. We also notice that some of the failure cases come from insecure grasps. This is probably due to sub-optimal grasp poses. Incorporating failure detection and re-planning capability could further mitigate this issue.

# I  NDF Training

We train per-category NDF models using 3D mesh models extracted from ShapeNet [60]. We adopt the same model architecture and learning hyperparameters as Simeonov et al. [8], namely a learning rate of 0.0001 and batch size of 16. The model is optimized using the Adam optimizer [61] and trained for 80k epochs. We employ 3D occupancy prediction as a pre-training task to acquire object descriptor features, and we randomly rotate and scale the object model to make the learned model more robust to shape variation. We use the same NDF models across all experiments in simulation. For real-world experiments, we further augment the training data by synthesizing partial point cloud to reflect the real-world perception input.

## J  LIBERO Qualitative Results and Failure Modes

Fig. 18 visualizes the execution of several LIBERO tasks. Typical failure modes of our approach include gripper collisions due to a tight cabinet drawers and object slippage due to sub-optimal grasp poses. Our system's performance is affected by the quality of the reference demonstrations, similar to other learning-from-demonstration methods. Thus, some of the failures can be improved through providing higher-quality demos. Additionally, incorporating the ability to replan would make the system more robust to skill execution failures.

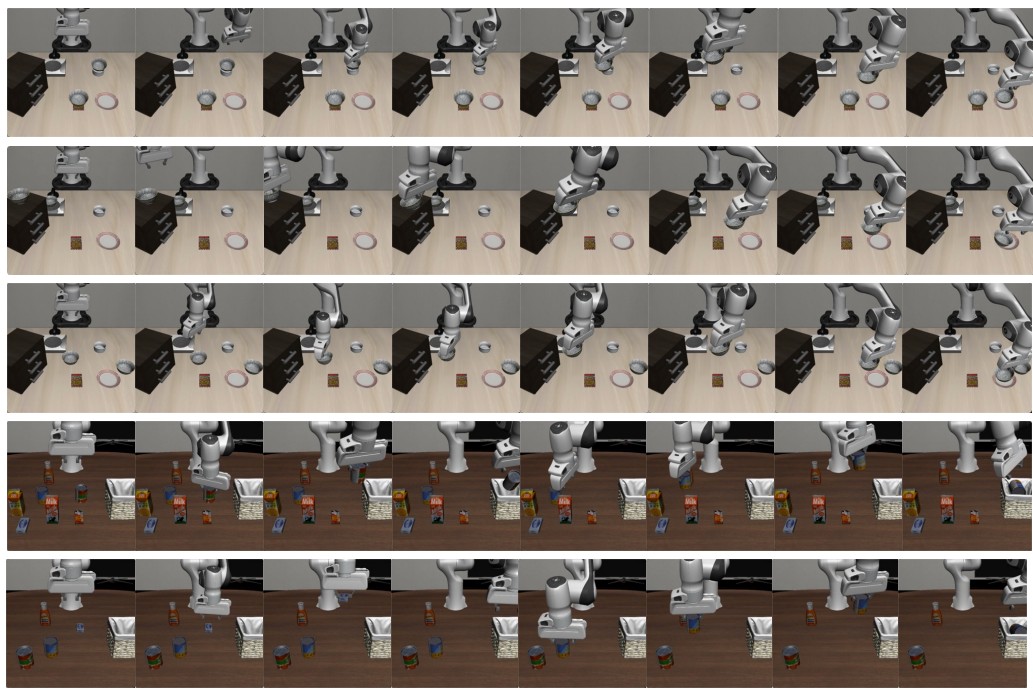

Figure 18: **LIBERO Results.** Key frames of three task execution processes for LIBERO benchmark.

# K  Pseudocode

Algorithm 1 shows the trajectory adaption process. Let $o$ and $o'$ be the source and target objects in the test scene for the adapted skill. Let $C_{acc}$ denote all the acquired constraints during the execution of prior skills (e.g., the grasp pose after executing PICK). Finally, let $T_w^o$ and $T_w^{o'}$ be poses for object $o$ and $o'$ respectively.

---

**Algorithm 1** Trajectory adaptation

---

**Declare:** Source object in test scene $o$, target object in test scene $o'$, robot end-effector $e$
**Declare:** Global accumulated constraints $C_{acc}$
**Declare:** Planned skills $\pi_* = \{d_1, ..., d_n\}$
1: **procedure** ADAPT-TRAJ($o, o', d$)
2:      $z_q, \mathcal{Z}_\tau \leftarrow d$
3:      $P_o \leftarrow$ PERCEPTION($o$)
4:      $P_{o'} \leftarrow$ PERCEPTION($o'$)
5:      $T_w^q \leftarrow$ NDF-OPTIMIZE($P_{o'}, z_q$)
6:                                         ▷ *Adapt query pose to test scene based on target object*
7:      $T_{o'}^q \leftarrow (T_w^{o'})^{-1} \cdot T_w^q$
8:      **if** $d.mode$=obj-obj **then**
9:          $T_e^{o'} \leftarrow C_{acc}[\langle o', e\rangle]$                    ▷ Extract constraints from $C_{acc}$
10:     **for** $z \in \mathcal{Z}_\tau$ **do**
11:          $T_w^q \leftarrow$ NDF-OPTIMIZE($P_o, z$)
12:                            ▷ *Adapt motion to test scene based on source object*
13:          **if** $d.mode$=obj-obj **then**
14:              $T_w^e \leftarrow T_w^q \cdot (T_{o'}^q)^{-1} \cdot (T_e^{o'})^{-1}$
15:          **else**
16:              $T_w^e \leftarrow T_w^q$
17:          **yield** $T_w^e$                        ▷ Yield target to controller
18:     **if** $d.mode$=obj-obj **then**
19:          delete $C_{acc}[\langle o', e\rangle]$             ▷ Remove constraint from $C_{acc}$
20:     **else**
21:          $T_{o'}^o \leftarrow T_{o'}^q \cdot (T_w^q)^{-1} \cdot T_w^o$       ▷ Acquire the last constraint
22:          $C_{acc}[\langle o, o'\rangle] \leftarrow T_{o'}^o$           ▷ Append constraint to $C_{acc}$

---

Note that the object poses used in our equations are just intermediate variables that help bridge the desired transformations, therefore these object poses do not need to carry any actual meaning. Here, we explain how we use NDFs to estimate novel object $P_{new}$'s point cloud transform $T_{new}$ w.r.t. a given reference object $P_{ref}$ with pose $T_{ref}$. To do so, we simply define the rotation of $T$ as identity, and define translation as the mean of $P_{ref}$:

$$z_{ref} \leftarrow \psi_{\text{NDF}}(T_{ref} \mid P_{ref})$$
$$T_{new} \leftarrow \text{NDF-OPTIMIZE}(P_{new}, z_{ref}).$$

The overall NOD-TAMP planning algorithm is shown in Algorithm 2.

**Algorithm 2** NOD-TAMP planner

**Declare:** Plan skeleton $[\hat{\pi}_1, \hat{\pi}_2, ..., \hat{\pi}_H]$
**Declare:** Task goal specification $Z_g$
1: **procedure** PLAN-NDF-SKILLS($[\hat{\pi}_1, \hat{\pi}_2, ..., \hat{\pi}_H], Z_g$)
2:     $\mathcal{D} \leftarrow [\,]$                                                          ▷ List of demos per skill
3:     **for** $i \in [1, ..., H]$ **do**
4:         $\mathcal{D} \leftarrow \mathcal{D} + [\{\tau\}_i]$, where $\{\tau\}_i$ is the trajectory set of skill $\hat{\pi}_i$
5:     $\pi_* \leftarrow$ **None**                                                      ▷ Optimal trajectory plan
6:     $c_* \leftarrow \infty$                                                              ▷ Lowest cost
7:     **for** $\pi \in$ PRODUCT($\mathcal{D}$) **do**                          ▷ All valid traj. sequences
8:         $c \leftarrow 0$                                                                  ▷ Feature cost
9:         $Z_{\text{acc}} \leftarrow \{\}$                                              ▷ All accumulated constraints
10:         **for** $i \in [1, ..., H-1]$ **do**
11:             $z_{\text{pre}}^i, z_{\text{eff}}^i \leftarrow$ PARSE($\pi[i]$)         ▷ Parse pre. and eff. constraints
12:             $z_{\text{pre}}^{i+1}, z_{\text{eff}}^{i+1} \leftarrow$ PARSE($\pi[i+1]$)
13:             $c \leftarrow c + ||z_{\text{eff}}^i - z_{\text{pre}}^{i+1}||$
14:                                                                                             ▷ Compute feature distance among skills
15:             $Z_{\text{acc}} \leftarrow Z_{\text{acc}} \cup \{z_{\text{eff}}^i\}$
16:                                                                                             ▷ Update acquired constraints
17:         **for** $\langle k, z_k \rangle \in Z_g$ **do**                          ▷ Enumerate goal constraints
18:             $\hat{z}_k \leftarrow Z_{\text{acc}}[k]$
19:             $c \leftarrow c + ||\hat{z}_k - z_k||$
20:                                                                                             ▷ Compute feature distance of the goal configuration
21:         **if** $c < c_*$ **then**                                                  ▷ Update best plan
22:             $\pi_* \leftarrow \pi; c_* \leftarrow c$
23:     **return** $\pi_*$

