# OpenReview forum: "NOD-TAMP: Generalizable Long-Horizon Planning with Neural Object Descriptors"
_robot-learning.org/CoRL/2024/Conference — CoRL 2024_

### Official Review · Reviewer_Stau · 2024-07-06
**Interesting work but many open questions**

**Originality:** 2
**Technical Quality:** 2
**Clarity Of Presentation:** 3
**Potential Impact:** 3
**Recommendation:** 3
**Confidence:** 4

**Review:**

The paper is well written and clear and the resulting robot experiments are impressive. However, I have some major concerns about this paper which need to be addressed.

1. I am not convinced by the usefulness of using neural descriptor fields to generate trajectories. You already have the start and goal end-effector poses, why not use classical trajectory generation with collision checking which is well established and shown to be fast and robust? This would also make it easier to implement your proposed future work to apply constraints to the trajectory such as keep the mug upright. I don't see any benefit from using learned trajectories for these simple pick and place tasks.

2. It is impressive that learned skills can be deployed on a real robot with as few as one demonstration. However, the specificity of of each skill makes this a much simpler problem. Separate skills for picking a bowl vs plate vs screw driver is in fact quite a limitation which I think should be addressed in the paper.

3. In the skill planning module I intuitively feel there may become ambiguities in selecting the right skill based solely on pointcloud and end-effector pose. Have the authors encountered any examples of this? What are the failure modes of the skill planning?

**Edit: Thank you to the authors for the additional explanations. On the basis of their responses I change my recommendation to Weak Accept.**

**Quality Of The Limitations Section:**

2

**Questions For Rebuttal:**

1. Can you justify the use of NDF for generating trajectories?
2. Can you discuss how the current formulation of the method requires very specific skills which limits the generalization of the method.
3. Have you encountered any ambiguities in optimizing the order of skills or are there any limitations to this method of chaining skills?

**Robotics Focus:**

4

**Summary Of Paper:**

This method uses an object centric TAMP approach to efficiently learn several tasks from very few demonstrations. They use neural descriptor feilds to encode joint object pointclouds with end effector poses. Then then use these encodings to generate trajectories and select which skills to chain together.

**Summary Of Recommendation:**

I am leaning reject because on the main contributions (generating trajectories from learned descriptors) is not sufficiently motivated.

---

### Official Review · Reviewer_s2BX · 2024-07-19
**Use neural descriptor fields to represent skill demonstrations**

**Originality:** 3
**Technical Quality:** 2
**Clarity Of Presentation:** 3
**Potential Impact:** 3
**Recommendation:** 3
**Confidence:** 4

**Review:**

The overall structure of the paper is good, however, the details of the method are not very clear; I understood the general idea behind the paper by making some educated guesses.

I think using NDFs to transform the demonstrations into a latent representation is a promising direction to learn skills from various data sources, and/or generalize to novel settings. I appreciate that there are a lot of experiments, but it is not very obvious in which cases the method shines and fails. For instance, the performance on MugPicking (0.85) is less than MugInsertion (0.90), which I would expect the reverse, and which is actually the case for other baselines except O/MP. And this is the general theme I felt while going over the experiments; I got that the method works, but could not imagine where and why the method would work or fail, or how difficult the task is to understand the increment over the baselines. For instance, does grasping a mug generalize to grasping a tumbler mug without a handle, or a teapot? This would help future readers to assess its current limitations and improve upon it.

Below are my other comments:
- In Sec. 3.2, it’s a bit vague whether ‘pre’ and ‘eff’ are normal point clouds or latent representations through NDFs. Is ‘param’ a latent representation as well? It would be nice to write this part with the notation consistent with Sec 3.1.
- Figure 2 could be improved. For instance, it was not obvious to me that those boxes in ‘skill planning’ are skills; I thought they were some sort of features.
- It’s hard to follow Sec 4.1. “Our skill adaptation module (1) transforms the skill trajectories to constraint-centric NDF feature trajectories and (2) adapts the trajectory to the observed scene via sequential optimization.” It’s not clear what is meant by the constraint-centric NDF feature trajectories. In line with my previous comment, it would be much easier to understand it with a proper notation.
- In Table 1, why does O/SR perform so well? Isn’t it basically picking random reference trajectories? Does this mean that the tasks do not require reasoning? I didn’t get why a random trajectory selection performs so well.
- In Table 2, while it’s mentioned that NOD-TAMP achieves strong performance, its ablated version NSC (without motion planning and skill reasoning) performs quite well, too!
- I am not sure if this can be thought of as long-horizon planning, but I agree that using neural descriptors might be good for learning behaviors from demonstrations that can generalize to other objects etc.

**Quality Of The Limitations Section:**

1

**Questions For Rebuttal:**

- Why does O/SR perform so well?
- What are the limitations of the method in addition to the mentioned computational bottleneck?

**Robotics Focus:**

4

**Summary Of Paper:**

This work leverages neural descriptor fields (NDFs) to generalize a skill demonstration to novel task settings (e.g., new object types, poses). A skill-reasoning module chains a sequence of skills together to solve a long-horizon task by checking the preconditions and effects of the skills in the latent representation provided by the NDF. Experiments done on a benchmark suite and a real-world robot show promising results.

**Summary Of Recommendation:**

This paper shows a novel way of learning from demonstrations using neural descriptor fields. The method uses NDFs to combine trajectories to solve long-horizon tasks. Although there are some vague points regarding the method, I think the set of experiments shows some promising results, which might ignite some future ideas.

---

### Official Review · Reviewer_3zeS · 2024-07-23

**Originality:** 2
**Technical Quality:** 4
**Clarity Of Presentation:** 4
**Potential Impact:** 2
**Recommendation:** 2
**Confidence:** 5

**Review:**

# Quality

The paper is of reasonably high quality. The motivation is clear, and the proposed method is sound (despite its limitations). The experiments are diverse and thorough, comprising both  simulated and real-world tasks. Ablations are thorough as well. The major quality issue is with the method design itself - see weakness section for more details. Additionally, I’d argue that their proposed “TAMP” section barely qualifies as TAMP, as the rough plan + constraints/effects have already been constructed - it’s mostly the “MP” part of TAMP, and jointly optimizing these two things is usually the hardest part.

# Clarity

This work is reasonably clear, although some of the trajectory adaptation math / intuition is unclear and I had to refer to the appendix algorithms to understand. It might make sense to move one or both of the algorithms to the main text.

# Originality

The papers is not particularly novel, as it adapts a well-explored framework (NOD / NDF) for efficiently representing geometric relationships to a rather restricted long-horizon setting using simple escape hatches to the challenging problems (multimodality+free-space motion -> motion planning; exponential search space -> tiny # of demonstrations).

# Significance

Because it only offers limited potential to generalize, and requires many prerequisites (rough plan, prespecified effects/preconditions), the paper is not particularly significant.

# Strengths

* The paper is well-written, and evaluates their method thoroughly on many different tasks.
* The sample-efficiency of the method is attractive, given the known-good properties of the NDF submodules.
* Structural generalization over SE(3) is a good property to have / exploit.
* The method outperforms the baselines.

# Weaknesses

* The major weakness of this paper is that it makes extremely significant assumptions about problem structure:
    * Task planning is already complete - a high-level plan is already provided as input, as are various constraints/effects for each transition. This is an extremely strong assumption - there is no real path given for generating these plans, which is highly nontrivial. In my opinion, this method doesn’t really qualify as TAMP because the task planning is provided.
    * The problems are not multimodal
    * The actual intermediate trajectories between keyframes are inconsequential
    * Each demonstration can be carved up into clear quasistatic keyframes
    * All objects are rigid, as are the attachments to the gripper
* The strong sample complexity seems like a red herring - as the algorithm itself fundamentally does not scale to larger sample complexity of varying quality / diversity. This is evident from the inclusion of a cartesian product in the algorithm for searching over trajectories, which is naive and exponential (and in fact, is usually the hardest part of long-horizon planning problems).
* There is no real discussion of multimodality, or generalizability to different quantities of objects (i.e. train on 0-2 mugs in the scene, test on a scene w/ 3 mugs)
* Composition of estimations (mug-object and object-object relationships) can incur various errors which may compromise the success rates over time

**Quality Of The Limitations Section:**

2

**Questions For Rebuttal:**

* How would this method scale to larger numbers of demonstrations?
* How can the task plans themselves be generated?
* Has this method been applied to truly long-horizon tasks? AKA moving 10+ objects around the scene.
* How can multimodality be handled?
* How can situations where the trajectory matters (aka “thread the needle” tasks) over the course of the entire trajectory be handled?

**Robotics Focus:**

4

**Summary Of Paper:**

In this work, the authors consider the task of long-horizon object rearrangement manipulation tasks that may involve multiple sequential rearrangments. The authors seek to combine two prominent frameworks: Neural Object Descriptors (which have been shown to promote intra-class and configuration generalization) and Task and Motion Planning (which is very useful for dealing with long-horizon planning). They devise an algorithm that learns NDF trajectories from demonstrations, and at test time searches over candidate trajectory options to find a plan to execute. They show promising results on several benchmarks.

**Summary Of Recommendation:**

In short, the execution and evaluation of the method are strong, but the method itself requires a large number of somewhat-unreasonable assumptions, and has limited utility without further exploration. There is no clear path forward to generalizability in new scenes which are not simple rearrangements (with minor in-class geometric variations) of previous demonstrations. Some of the fundamental limitations ought to be addressed - otherwise I see this contribtuition as incremental.

---

### Author Rebuttal · Authors · 2024-08-09

We thank the reviewers for their time and constructive suggestions. We are grateful for their positive recognition of our work’s high quality and clear motivation (3zeS), good originality (s2BX, uZpd), and impressive performance with thorough evaluation (3zeS, Stau). We address each reviewer’s comments in individual response and a revised manuscript. But first, we emphasize the significance of our work by contextualizing it within current trends in imitation learning, particularly Behavior Cloning (BC).
Typical BC methods require extensive teleoperation data [1, 2] and struggle with variations in shape, scene layouts, lighting, and camera views. For example, OpenXEmbodiment [2] includes ~800k demos but is largely limited to short pick-and-place tasks. In contrast, our system acquires manipulation policies from very few human demos and generalizes well across object shapes, poses, scenes, and long-horizon goals. We summarize tasks from our paper that highlight these capabilities at the end of this response.
While end-to-end BC is popular, we believe alternative approaches with different trade-offs in sample efficiency and assumptions are also of great interest to the community. Drawing from classical Task and Motion Planning (TAMP) systems, we developed a new system that generalizes across various long-horizon tasks. Unlike classical TAMP planners, our system is more flexible, doesn't require accurate environment models, handles non-prehensile contact, and avoids hand-engineered manipulation skills.
| Domains  | Task Name | Attributes  |
|:---:|:---:|:---:|
| Real-world (Fig. 4) | Insert Screwdriver  | Tight-tolerance |
|  | Make Coffee | Long-horizon; Tight-tolerance; Articulated Objects |
|  | Sort Tableware | Long-horizon; Skill reuse |
|  | Clear Mugs (Same/Multi Goal)  | Long-horizon; Skill reuse; Geometric reasoning: grasp different parts of the mug to achieve different placements |
|  | Use Tool | Long-horizon; Skill reuse; Geometric reasoning: different grasping strategies to use the tool; Non-prehensile: use the tool to operate blocks |
| Simulation (Fig. 3) | Mug Insertion  | Tight-tolerance |
|  | Table Clear | Long-horizon; Skill reuse |
|  | Table Clear Hard | Long-horizon; Skill reuse; Geometric reasoning |
|  | Tool Hang | Long-horizon; Tight-tolerance |

[1] DROID: A Large-Scale In-the-Wild Robot Manipulation Dataset, Khazatsky et al., arXiv 2024;

[2] Open X-Embodiment: Robotic Learning Datasets and RT-X Models, O’Neill et al., ICRA 2024.

---

### Decision · Program_Chairs · 2024-09-04

**Decision:**

Accept

**Comment:**

The reviewers raised concerns regarding the fact that a plan skeleton is provided, so the task planning is essentially given, lack of implementation details, unstated assumptions. The rebuttal submitted by the authors addressed these concerns to a large extent, and two out of three reviewers recommend acceptance.